# Clinical Safety and Tolerability of A2NTX, a Novel Low-Molecular-Weight Neurotoxin Derived from Botulinum Neurotoxin Subtype A2, in Comparison with Subtype A1 Toxins

**DOI:** 10.3390/toxins13110824

**Published:** 2021-11-22

**Authors:** Toshiaki Takeuchi, Tsuyoshi Okuno, Ai Miyashiro, Tomoko Kohda, Ryosuke Miyamoto, Yuishin Izumi, Shunji Kozaki, Ryuji Kaji

**Affiliations:** 1Department of Clinical Neuroscience, Graduate School of Medicine, Tokushima University, Tokushima 770-8503, Japan; c201356007@tokushima-u.ac.jp (T.T.); no_license_doctor@yahoo.co.jp (T.O.); amiyashiro818@yahoo.co.jp (A.M.); ninigii@gmail.com (R.M.); yizumi@tokushima-u.ac.jp (Y.I.); 2Department of Veterinary Sciences, School of Life and Environmental Sciences, Osaka Prefecture University, Osaka 598-8531, Japan; kohda@vet.osakafu-u.ac.jp (T.K.); kozaki@center.osakafu-u.ac.jp (S.K.)

**Keywords:** botulinum toxin, subtype A2, clinical tolerability, safety, patients

## Abstract

All the botulinum type A neurotoxins available for clinical use are of the A1 subtype. We developed a subtype A2 low-molecular-weight (150 kD (kilo Dalton)) neurotoxin (A2NTX) with less spread and faster entry into the motor nerve terminal than A1 in vitro and in vivo. Preliminary clinical studies showed that its efficacy is superior to A1 toxins. We conducted an open study exploring its safety and tolerability profile in comparison with A1LL (LL type A1 toxin, or onabotulinumtoxinA) and a low-molecular-weight (150 kD) A1 neurotoxin (A1NTX). Those who had been using A1LL (*n* = 90; 50–360 mouse LD50 units) or A1NTX (*n* = 30; 50–580 units) were switched to A2NTX (*n* = 120; 25–600 units) from 2010 to 2018 (number of sessions ~27, cumulative doses ~11,640 units per patient). The adverse events for A2NTX included weakness (*n* = 1, ascribed to alcoholic polyneuropathy), dysphagia (1), local weakness (4), and spread to other muscles (1), whereas those for A1LL or A1NTX comprised weakness (*n* = 2, A1NTX), dysphagia (8), ptosis (6), local weakness (7), and spread to other muscles (15). After injections, 89 out of 120 patients preferred A2NTX to A1 for the successive sessions. The present study demonstrated that A2NTX had clinical safety up to the dose of 500 units and was well tolerated compared to A1 toxins.

## 1. Introduction

Botulinum neurotoxins (BoNT) are known as the most potent biological substance to cause risk of death [1,2]. Many attempts to use BoNT as a biological weapon were made in the past [3,4,5]. In the US Army, investigators of Camp Detrick conducted research on BoNT from 1943 to 1956, to develop toxins and effective vaccines during and after World War II [4,6,7]. The isolation and purification of BoNT originated from the Hall strain of *Clostridium botulinum* [8,9], which made it possible to produce botulinum toxins and toxoid vaccines at a large scale [6,7]. In the 1960s, when the treaty for biological weapons banned wartime use of BoNTs, Dr. Edward J. Schantz, a basic scientist who had worked at Camp Detrick, supplied the toxin for scientific purposes [10,11,12]. Dr. Alan B. Scott, an ophthalmologist, came across an idea of its clinical use in small doses to reduce muscle hyperactivity [13,14]. He first tested *botulinum toxin* type A (BoNT/A) from the Hall strain provided by Dr Schantz in humans in 1978, after he received permission from the FDA [14]. Ten years later, Allergan Inc. acquired the rights to distribute the drug or onabotulinumtoxinA (BOTOX^®^). The company extended the clinical research to obtain FDA approval for its use in dystonia, spasticity, migraines, and others [15,16].

Botulinum neurotoxins (BoNTs) are categorized into seven immunologically distinct serotypes, BoNT/A to/G (subtypes A-G). BoNT/A, which exist in various molecular weights, LL (900 kD), L (500 kD), M (300 kD), and S (150 kD) [17,18,19]. LL and L toxins contain a hemagglutinin (HA) component and an NTNH (non-toxin, non-hemagglutinin) component, which are not essential for the action of neurotoxins (S toxin), although LL toxin may have advantages over S because of its lower diffusibility, due to its larger molecular weight [19,20,21]. M toxin is composed of NTNH and S toxin [19,22,23]. Neurotoxin (NTX), or S toxin, comprises the heavy chain (HC), which has binding and translocation domains, and the light chain (LC), which has a catalytic domain [19,22,24].

Detailed research on amino acid sequences in each serotype revealed subtypes among type A toxins (A1, A2) [25,26,27]. These subtypes have been increasingly recognized on each serotype [27,28,29]. Serotype A is now divided into subtypes A1-A8, and only A1 toxin from the Hall strain (onabotulinumtoxinA or BOTOX^®^, abobotulinumtoxinA or Dysport^®^, incobotulinumtoxinA or Xeomin^®^) is clinically available in the US [30,31], except for type B toxin (rimabotulinumtoxinB or Myobloc^®^/Neurobloc^®^). Type F was once used for clinical research but was found to have a shorter duration of action than type A [32,33]. There has, however, been a safety concern for subtype A1 neurotoxins in treating cerebral palsy in children and dystonia, because culminating cases of death were reported, possibly due to their spread to respiratory muscles [1,34].

Sakaguchi and colleagues [35] have obtained unique strains of *C. botulinum* from cases of infant botulism in Japan (Chiba-H and Kyoto-F), and they found that one of these strains (Chiba-H) only produces M-sized type A toxins, which are free from hemagglutinin. The NTX produced from these Japanese strains shares 89% amino acid sequence homology with that derived from the A1 subtype (A1NTX), which was eventually categorized as subtype A2 [36]. Because of the high yield and purity of the M toxin produced by the Chiba-H strain, which had already been cleaved by endogenous proteases needed for the A1 subtype, it was possible to convert the M toxin into highly purified S toxin (botulinum neurotoxin or BoNT) for clinical use (A2NTX). The latter was found to be less diffusible [26,37] and more efficacious per mouse LD50 unit in vitro and in vivo [37], but its specific toxicity per mouse LD_50_ in vitro is similar to A1NTX [38,39].

The first-in-man clinical study [40] indicated that A2NTX is around 1.54 times as potent as the same unit of onabotulinumtoxinA (A1LL), whereas 6.5 mouse LD50 units of A2NTX and 10 mouse units of A1LL showed comparable reduction in and duration of compound muscle action potential (CMAP) after being injected into the extensor digitorum brevis (EDB) muscle of the foot, with less spread to adjacent muscles (Abductor Hallucis muscle).

In a small-sized (*n* = 31), double-blind, controlled study of head-to-head comparison of onabotulinumtoxinA (A1LL) and A2NTX [41], 300 u of each subtype was injected into the calf muscles of post-stroke spasticity patients. A2NTX showed a significantly higher reduction in spasticity (measured by modified Ashworth scale) 30 days after injection, and less spread of the effect, as measured by the hand grip of the unaffected side, than A1LL. The functional independence measure (FIM) was also significantly improved for A2NTX, but not for A1LL, suggesting higher efficacy and safety of A2.

Here, we report a long-term clinical study of A2NTX, exploring its safety and tolerability in comparison with A1LL and A1 S toxins (A1NTX). A part of this study [41] was registered (ClinicalTrials.gov identifier: NCT01910363), and was published or reported as abstracts or proceedings [41,42,43]. The entire protocol was approved by the IRB of Tokushima University, Japan (date of approval: 31 March 2004, No. 2005–216, with revisions in 2006 and 2010). The study was conducted according to the guidelines of the Declaration of Helsinki. The procedure of administering to human subjects was fully legitimate until 2018. Informed consent was obtained from all the patients in a written form.

## 2. Results

### 2.1. Patients

The patients who had A1LL (onabotulinumtoxinA), A1NTX (S-toxin from the A1 subtype), and A2NTX (S-toxin from the A2 subtype) were diagnosed as having spasticity or dystonia/tremor. The total number of patients for each injection group was 90 (34 female, 56 male) for A1LL, 89 (23 female, 66 male) for A1NTX, and 120 for A2NTX (53 female, 67 male) (Table 1), which included patients with spasticity (21 for A1LL, 29 for A1NTX, and 42 for A2NTX) and dystonia (69 for A1LL, 60 for A1NTX, and 78 for A2NTX). The breakdown of spasticity patients was as follows: post-stroke (17 for A1LL, 10 for A1NTX, and 25 for A2NTX), hereditary spastic paraplegia (HSP) (2 for A1LL, 11 for A1NTX, and 12 for A2NTX), acquired spinal cord injuries (1 for A1LL, 4 for A1NTX, and 4 for A2NTX), and others, including cerebral palsy. Because of their promise in efficacy, A1NTX (*n* = 9) and A2NTX (*n* = 28) were also used for generalized and other dystonias or tremors involving larger parts of the body than cervical dystonia or blepharospasm. The maximum dose was 600 units of A2NTX in one patient with generalized dystonia, who eventually chose surgery (deep brain stimulation), whereas the rest of the doses were 500 units of A2NTX (Figure 1A,B). The maximum dose per kg body weight was 13.54 units in a subject who received 600 units, and was up to 11.57 units in one subject who had received 500 units per session. The ceiling of 500 units for A2 seemed to be appropriate because two patients who underwent injections of 580 units of A1NTX developed generalized weakness (vide infra). Those who switched from A1LL to A2NTX included three patients who became unresponsive to A1LL, as diagnosed by frontalis test [44] (secondary non-responders).

As for the patients who had A2NTX, the age range was 14–93 years (mean: 54.5 years), the duration of treatment was 1–8 years (mean: 5.15 years), the cumulative dose was 25–11,640 units (mean: 2527 units), and the number of injection sessions was 1–27 (mean: 9.5). In the A1NTX-injected group, the age range was 19–79 years (mean: 53 years), the duration of treatment was 1–10 years (mean: 3.0 years), the cumulative dose was 50–8200 units (mean: 2653 units), and the number of injection sessions was 1–25 (mean: 9.0). Overall, the patients’ background and dosing did not differ significantly between the A1NTX- and A2NTX-treated groups. The cumulative doses and ranges could not be analyzed for the A1LL patients because many had been referred to the study from other clinics, where the previously used doses were unknown.

### 2.2. Adverse Events

Table 1 summarizes the adverse events among the A1LL-, A1NTX-, and A2NTX-treated groups.

The patients were asked about any adverse events at each visit by a nurse or clinical research coordinator (CRC) who was blinded to the dose and the preparation of BoNT (A1LL, A1NTX, A2NTX), and physicians examined muscle power manually at each visit, not only of the injected, but also the un-injected muscles. The results were analyzed according to ICH E2A (Clinical Safety Data Management: Definitions and Standards for Expedited Reporting) [45] and categorized as follows:

Mild: an event that is easily tolerated by the participant, causing minimal discomfort and not interfering with everyday activities.

Moderate: an event that causes sufficient discomfort and interferes with normal everyday activities.

Severe: an event that prevents normal everyday activities and may require hospitalization.

Serious: any untoward medical occurrence that, at any dose, results in death, is life-threatening, requires inpatient hospitalization or prolongation of existing hospitalization, or results in persistent or significant disability/incapacity.

When serious adverse events, or any of the adverse events in question, arose, the independent committee for safety was called and members were asked to assess whether the event was causal or not.

Since all the complete medical records of the A1LL- or A1NTX-treated groups were not available, due to their observation period (A1NTX: 2006–2010) or referral from other medical institutions (A1LL), a thorough review of any adverse events was limited for A1 toxin recipients, and the bias was in favor of A1.

### 2.3. Mild Adverse Events

Routine blood samples showed no major changes after the initial dosing of 50 units, except for three patients whose liver function data (AST, ALT, γ-GTP) deteriorated slightly. This was the only mild adverse effect, except for local pain or ecchymosis.

### 2.4. Moderate Adverse Events

Patients were specifically asked about and checked for the presence of unwanted weakness in the injected muscle (local weakness), in the un-injected neighboring muscles (spread to other muscles), in the muscles of distant body parts (generalized weakness), dysphagia (in the case of cervical dystonia), and ptosis (in the case of blepharospasm). Table 1 depicts that the incidence of these adverse events was higher in the A1NTX-injected group than in A2NTX. Despite being injected with smaller doses, those who had A1LL also showed higher incidence of these parameters than A2.

### 2.5. Serious Adverse Events

There were three cases with serious adverse events of generalized muscle weakness; there was one from the A2NTX group (Table 2) and two from A1NTX.

Case A2–53 was a 60-year-old man who had a right putaminal hemorrhage 5 years prior to the dosing of 400 units of A2NTX into the left upper and lower extremity muscles. He had been diagnosed as having alcoholism, and his food intake was decreased 1 month before the dosing. He developed weakness in his lower limbs and became unable to walk 2 weeks after the dosing. The neurological findings were compatible with subacute alcoholic polyneuropathy, and he recovered fully after vitamin B1 supplements. Repetitive motor nerve stimulation did not show waxing at 30 Hz, typically observed in botulism. The report of the independent safety committee was that the event was not causal to the dosing.

The other case (A1–13) was a 44-year-old woman with generalized dystonia, who developed generalized weakness in her limbs after two repeated injections of 580 units of A1NTX into the lower limb muscles at 10 week intervals. She became unable to walk and was hospitalized for 4 weeks, but made a full recovery. Electrophysiological studies showed increased jitter using single-fiber EMG, and more than a 30% incremental response (waxing) after 30 Hz motor nerve stimulation in the hand muscle, suggesting the effect of the botulinum toxin in distant muscles. The safety committee judged this case as having a side effect.

Another case of cerebral palsy (case A1–42), a 42-year-old woman, developed generalized weakness, including hand grip at 2 weeks after having injections of A1NTX (580 units) into the lower limb muscles. She remained ambulatory and the weakness recovered completely within 8 weeks.

Table 2 depicts the rest of the adverse events in the A2NTX-treated group, collected from all the medical records for more than 8 years (2010–2018), by a CRC or a physician blinded to the dosing schedule. Falls were reported in three cases. One case (case 50) had a hip bone fracture at the peak effect of A2NTX injected into the leg (4 weeks after the injection), which was judged as causal because the patient became ambulatory and had the chance to walk for a long distance after a period of inactivity. This patient recovered completely after replacement of the joint. There were two deaths (both with generalized dystonia), which were judged to be not directly related to the injection, one from a suicide and the other from an accident. Two patients had relapses of stroke, not immediately following the injection (>2 weeks).

### 2.6. Secondary Non-Responders to A1LL

Those patients with resistance to A1LL, tested by a frontalis test [44], included a 46-year-old man (case A2–62), a 35-year-old woman (case A2–89), and a 35-year-old woman (case A2–91), all suffering from cervical dystonia. All of them, who had been treated with 240 units of A1LL with no benefits, clinically responded to 400–500 units of A2NTX injections. The frontalis test in one patient (A2–91) regained asymmetry of the frontal crease, suggesting a clinical response when using 50 units of A2NTX.

### 2.7. Patients’ Predilection

Eighty-nine of the 120 patients (74%), who had been treated with A1LL or A1NTX at intervals of 8–12 weeks, eventually chose A2NTX injections, and the intervals became 10–24 weeks. Nine patients (7.5%) switched back to A1LL injections at local neurologists because they became unable to visit. Twenty-two (18.3%) discontinued BoNT injections because they recovered fully or attained a plateau of clinical benefits.

### 2.8. Secondary Non-Responders to A2NTX

No individuals became unresponsive to A2NTX during our observation period.

## 3. Discussion

The present study was the first to report the long-term tolerability and safety of A2NTX, and showed that it was well tolerated up to the dose of 500 units for spasticity and dystonia. The limitation was that it was a retrospective case-control study that was not designed for statistical comparison between the preparations.

There have been several experimental studies in accordance with the present clinical results. An in vitro study showed that A2 enters neuronal cells faster than A1 [26]. Measuring grip strength in rats, A1 spreads weakness more readily to the contralateral un-injected limb than A2 [37,46]. In the past, 1 unit of botulinum neurotoxin has been measured as a dose that is lethal in mice at 50% probability after intraperitoneal injection. LD50 values after intramuscular injection were studied in mice [47], and it was shown that those for A1LL and A2NTX were 93 and 166 U/kg, respectively. Since the intraperitoneal route is readily borne to systemic circulation compared to the local intramuscular route, the same units of A2NTX are safer than those of A1LL, if injected locally, as in the case of clinical settings. These results in mice and rats were replicated in a primate model [48]. The spread of the effects of the toxin was shown, at least partially, through trans-neuronal pathways, and was much less for A2 than A1 in an immunohistochemical study [49].

Although the present study was not designed to compare the incidence statistically, adverse effects due to the spread of A2NTX were less than those of A1NTX, and were even less than those who had smaller doses of A1LL. If these are confirmed, the margins of safety would be even higher, considering that 1 unit of A2 is 1.54 times as effective as 1 unit of A1 in humans [40].

With regards to the patients’ predilection, A2NTX was favored over A1 toxins and was even useful in secondary non-responders of cervical dystonia to A1LL if used at relatively high doses (400–500 units) of A2.

Detailed analyses of the adverse events revealed three cases with falls, which is an obvious underestimate, because of the disability, the age, and the number of subjects [50]. Bone fracture was reported in three cases. Relapses of stroke were observed in two cases, which was also small in number, because the stroke relapse rate in Japan is around 5% per annum [51]. One patient with alcoholism developed full-blown polyneuropathy with generalized weakness after A2NTX, which was reversed with vitamin B1 injections. It is important, however, that those with other potential risks of developing weakness should be carefully screened before the use of A2NTX, but this caution would apply to other preparations of BoNTs. As such, A2NTX at doses up to 500 units does not carry extra risks compared to other neurotoxins, as far as the current case series are concerned.

Concerning the treatment-related side effects (Table 1), the spread of weakness to other muscles was less for A2NTX than for A1NTX or A1LL. The large molecular weight of A1LL is possibly related to its immunogenicity [52], resulting in antibody development, but could be beneficial in limiting its diffusion to other muscles, locally or through the CNS [20,49]. Despite its low molecular weight (150 kD), A2NTX spread less than not only A1NTX, but also A1LL, despite the limitations of the present study. Our previous report also demonstrated the reduced spread of A2NTX compared to A1LL in lower limb post-stroke spasticity [41]. The exact reason for this is unknown, but it may be related to its higher affinity and faster binding to the receptor’s gangliosides/SV2, possibly because of the differences in the amino acid residues in the heavy chain [24,26,53]. If so, it is reasonable to observe its efficacy in secondary non-responders to A1LL, who could have antibodies to A1 toxins, since A2 might find its way to the receptor faster than the antibody in high doses. It is also conceivable that A1 and A2 have different immunogenicity [54].

## 4. Conclusions

Although the present retrospective case-control study was not designed for statistical comparison between the preparations, the current results underscored the safety and tolerability of A2NTX, a low-molecular-weight neurotoxin, up to 500 mouse LD50 units per 10–12 weeks, in the long-term condition, without eliciting secondary non-responders of its own.

## 5. Materials and Methods

### 5.1. Toxins

Neurotoxins (molecular weight: 150 k Dal) were produced and purified from Chiba-H strain of *Clostridium Botulinum*, isolated from honey associated with cases of infant botulism, for subtype A2 (A2NTX) and from strain 62A for subtype A1 (A1NTX) as described elsewhere [22]. The toxicity of purified neurotoxin A2NTX, titrated by serial 2-fold dilution intraperitoneal injection measured as a mean 50% lethal dose (LD50), was 5.2 × 10’ LD50/mg protein, which was nearly the same as that of 62A neurotoxin (A1NTX) (5.3 × 10’ LD50/mg protein). We also compared the safety of these small-molecule type A neurotoxins with A1LL or onabotulinumtoxinA (BOTOX^®^, Allergan Inc., Irvine, CA, USA), whose approved doses were up to 360 mouse LD50 units for spasticity in Japan (as of 2018, up to 400 u). Those preparations of A1NTX and A2NTX were stored in a deep freezer (<−70 °C) and thawed immediately before use.

### 5.2. Criteria for Entry

Patients with spasticity, dystonia or tremor, being treated with A1LL or A1NTX, whose responses fell short of the subject’s needs, or their benefits disappeared as judged by frontalis test (secondary non-responder). The ages of patients were between 10 and 95.

Written consent was obtained from all the subjects or their parents if they were under the age of 20, after giving full information on the exploratory nature of the study and the data being published for research purposes.

### 5.3. Clinical Test Periods

During the period of October 2006 to November 2010, we used A1NTX (up to 580 units) mainly for spasticity and dystonia. Since there had been no BoNTs available for spasticity in Japan, this study started as a proof-of-concept clinical study of A1NTX for spasticity. A2NTX was developed in 2010 after in vivo and in vitro studies [37,46]. The Institutional Review Board (IRB) of Tokushima University approved the use of A2NTX up to 600 mouse LD50 units for spasticity or dystonia in 2010 (approval number TU-214) after the first-in-man study using 6.5 units in the extensor digitorum brevis (EDB) muscle in healthy subjects (published in 2014) [40], and 300 units in post-stroke spasticity (published in 2015) [41]. The previous study showed the potency of A2NTX to be 1.54 times as much as that of A1LL in humans [40].

After A1LL (onabotulinumtoxinA) was registered for spasticity in Japan using doses up to 360 units in October 2010, those treated with A1LL who wished to try A2NTX were entered into the A2NTX study (*n* = 90) after informed consent at any time until March 2018. Those who had been treated with A1NTX from October 2006 until December 2018 (*n* = 89) were also allowed to switch to A2NTX or A1LL at any time after December 2010. A total of 120 patients received A2NTX injections (Figure 2).

### 5.4. Injection Protocol

All the subjects of A2NTX group had an initial dosing of 50 units, followed by 4–6 weeks with the second dosing of up to 400 units. The maximum doses were increased to 500 units in some spasticity patients who required further benefits from the injections (Figure 1). Injection session intervals thereafter ranged from 8 to 12 weeks until the plateau of clinical benefits was reached. After the second dosing, the patients were asked for their predilections on the BoNT preparations, and they were allowed to switch back to the original regimen (A1NTX or A1LL) at any time after the start of A2NTX dosing.

### 5.5. Adverse Events

Blood samples including those for serum electrolytes, liver, renal and hematological routines were obtained before and after the initial dosing of 50 units, as well as when they were needed. Suspected non-responders to BoNTs were tested with frontalis test [44], which evaluates the effect of BoNT by comparing the maximum voluntary eyebrow elevation on each side after injection of BoNT or saline on each side to the patients in a blinded fashion.

## Figures and Tables

**Figure 1 toxins-13-00824-f001:**
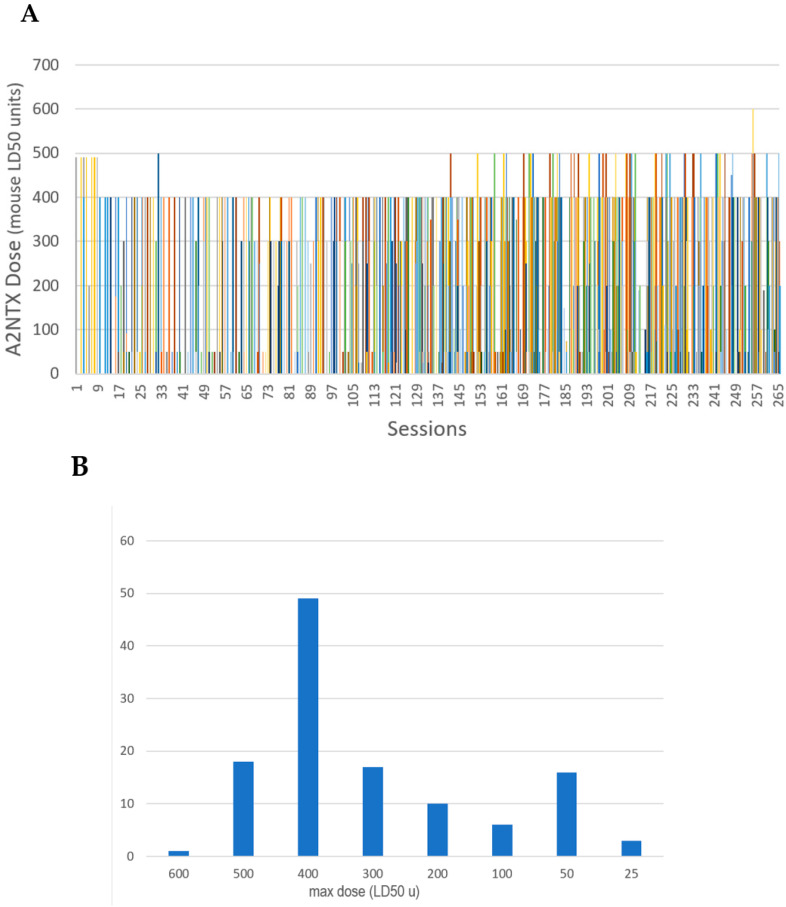
(**A**) Dosing of A2NTX among 120 subjects in all the injection sessions. Individuals are depicted with the same color; (**B**) number of subjects per dose range. Patients with spasticity or dystonia were treated with 400 units most frequently, but 500 units was also well tolerated.

**Figure 2 toxins-13-00824-f002:**
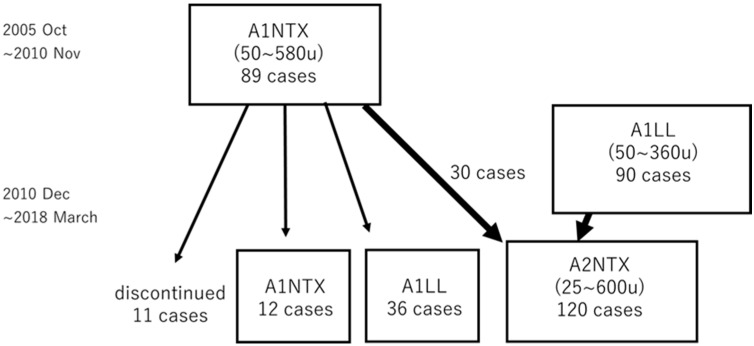
Flow chart of the subjects with respect to the toxin preparations. The first part from 2005 to 2010 included A1NTX. A1LL had been used for cervical dystonia and blepharospasm (max. dose 240 units) before 2010 in Japan and was approved for upper and lower limb spasticity in 2010 (max. dose 360 units). After IRB approval, those who had A1LL and A1NTX switched to A2NTX for the second part of the study.

**Table 1 toxins-13-00824-t001:** Cumulative occurrence of treatment-related side effects per individual among A1LL-, A1NTX- and A2NTX-injected groups.

	A1LL	A1NTX	A2NTX
No. of Total Subjects	91	89	120
Total Months of Observation (max)	50	50	87
**Spasticity** (dose)	300–360 u	290–580 u	300–500 u
number of subjects (age)	21 (28–96 y)	29 (28–79 y)	43 (41–96 y)
generalized weakness	**0**	**2 (6.9%)**	**1 (2.3%)**
local weakness	0	3 (10.3%)	3 (7.0%)
spread to other muscles	**3 (14.3%)**	**12 (41.4%)**	**1 (2.3%)**
**Cervical/truncal dystonia** (dose)	100–240 u	290–580 u	300–500 u
number of subjects (age)	19 (25–81 y)	30 (26–73 y)	31 (25–81 y)
local weakness	1 (5.3%)	3 (10.0%)	1 (3.2%)
dysphagia	**3 (15.8%)**	**5 (16.7%)**	**1 (3.2%)**
**Blepharospasm** (dose)	50 u	50 u	50–100 u
number of subjects (age)	51 (30–90 y)	21 (34–78 y)	19 (30–90 y)
ptosis	**4 (7.8%)**	**2 (9.5%)**	**0**
**Others** (including tremor and other dystonia)		29–580 u	25–500 u
number of subjects		9	28
spread to other muscles		**3**	**0**

**Table 2 toxins-13-00824-t002:** Summary of major adverse events in A2NTX-injected group.

Moderate	Severe	Serious
**falls**	**bone fracture**	**generalized weakness**
case 5	case 2	case 53
case 17	case 50 *	**relapses of stroke**
case 47 *	case 101	case 43
**aggravation of tremor**		case 118
case 11 *		**death**
		case 10 (suicide)
		case 35 (asphyxia in an accident)

* Considered as causal.

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
