# Peer review of "Clinical Safety and Tolerability of A2NTX, a Novel Low-Molecular-Weight Neurotoxin Derived from Botulinum Neurotoxin Subtype A2, in Comparison with Subtype A1 Toxins"

_toxins, 2021, doi:10.3390/toxins13110824_

Round 1

Reviewer 1 Report

Information on several issues were not sufficiently provided in introduction, including:

1. Advantages of A2NTX based on previous animal studies, particularly the advantages compared with A1NTX or A1LL, which are important to a better understand of the design and purpose of the study.

2. Important safety issues and side effects of clinical used botulinum neurotoxins, particularly those related to A1NTX or A1LL.

3. A summary of the reasons why A2NTX was proposed to be clinically used probably as an alternative to A1NTX or A1LL.

Besides, detailed dosing information should have been clearly provided, such as number of dosing and cumulative doses, in each group of patients. Data of average level and distribution in each group might be helpful to compare the effects of the toxins.

Author Response

We appreciate the reviewer for kind and helpful comments, and we revised mx accordingly.

1. Advantages of A2NTX based on previous animal studies, particularly the advantages compared with A1NTX or A1LL, which are important to a better understand of the design and purpose of the study.

- We added ' Because of the high yield and purity of the M toxin produced by Chiba-H strain, which had already been cleaved by endogenous proteases needed for A1 subtype, it was possible to convert the M toxin into highly purified S toxin (botulinum neurotoxin or BoNT) for clinical uses (A2NTX). The latter was found to be less diffusible [17,18] and more efficacious per mouse LD50 unit in vitro and in vivo [18], but its specific toxicity per mouse LD50 in vitro is similar to A1NTX [19,20]. ' at para 4, p.3.

2. Important safety issues and side effects of clinical used botulinum neurotoxins, particularly those related to A1NTX or A1LL.

 - We added at para 3 of p.3 'There has however been safety concern of these subtype A1 neurotoxins in treating cerebral palsy children, because cases were reported culminating in deaths possibly due to their spread to respiratory muscles [14].' . 

3. A summary of the reasons why A2NTX was proposed to be clinically used probably as an alternative to A1NTX or A1LL.

 -We revised at para 5, p.3  'The first-in-man clinical study indicated that A2NTX is around 1.54 times as potent as the same unit of A1LL with less spread to adjacent muscles [21]. Small-sized double blind controlled study of head-to-head comparison of onabotulinumtoxinA and A2NTX in post-stroke spasticity indicated higher efficacy and safety of A2 [22].'

Besides, detailed dosing information should have been clearly provided, such as number of dosing and cumulative doses, in each group of patients. Data of average level and distribution in each group might be helpful to compare the effects of the toxins.

We revised at para 1, p.5 'As for those who had A2NTX, the age was 14-93 years (mean 54.5 years), the duration of treatment 1-8 years (mean 5.15 years), the cumulative dose 25-11640 units (mean 2527 units), the number of injection sessions 1-27 (mean 9.5). In A1NTX-injected group, the age was 19-79 years (mean 53 years), the duration of treatment 1-10 years (mean 3.0 years), the cumulative dose 50-8200 units (mean 2653 units), the number of injection sessions 1-25 (mean 9.0). Overall, the patients’ background and dosing did not differ significantly between A1NTX- and A2NTX-treated groups. The cumulative doses and its range could not be analyzed for A1LL patients, because many of those had been referred to the study from other clinics, where the previously used doses were unknown. '

As for the cumulative doses for A1LL (BOTOX), most of the patients had been referred by other clinics, and detailed survey in the past medical record was not possible. 

Reviewer 2 Report

The authors developed a subtype A2 low molecular weight (150kD) neurotoxin (A2NTX), with less spread and faster entry into the motor nerve terminal than A1 in vitro and in vivo. They conducted an open study exploring its safety and tolerability profile in comparison with A1LL (onabotulinumtoxinA) and low molecular weight (150kD) A1 neurotoxin (A1NTX). Those who had been using A1LL (n=90; 50-360 mouse LD50 units) or A1NTX (n=30; 50-580 units) were switched to A2NTX (n=120; 25-600 units) from 2010 utill 2018 (number of sessions ~ 27, cumulative doses ~11,640 units per patient). The authors postulated that the present study demonstrated that A2NTX had the clinical safety up to the dose of 500 units, and was well tolerated compared to A1 toxins.

The results of the study are new. Nevertheless, this manuscript needs substantial improvements and corrections before publishing may be possible.

General points:

Please correct all spaces between the words and references numbers at the end of the sentences.

Please add to your manuscript List of Abbreviations.

Please change your title to:  Clinical Safety and Tolerability of A2NTX, a Novel Low Molecular Weight Neurotoxin Derived from Botulinum Neurotoxin Sub-type A2, In Comparison with Subtype A1 Toxins.

Special points:

My big problem with this study that this Neurotoxin Sub-type A2 is not authorized for patients, but the authors did these experiments?  Please describe very exactly in which countries is this medication authorized and possible?

Is this study an official firs human study for this medication?

All you arguments is only your personal opinion, but without scientific justification. Please describe very exactly how you did analysing of the efficiency and side effects in patients?

Keywords: please add also to keywords: patients

Introduction

First of all, please describe in your Introduction section: is this Neurotoxin Sub-type A2 authorised for patient’s treatment? In which countries exactly is this Neurotoxin Sub-type A2 authorized for patients?

You said: Botulinum neurotoxins (BoNT) are known as the most potent biological substance to risk people to death. Many attempts to use BoNT as a biological weapon were made in the past.

Please add multiple references at the end of this sentence.

You said: In US Army, investigators of Camp Detrick conducted researches on BoNT from 1943 to 1956 to develop toxins and effective vaccines during and after World War II. The isolation and purification for BoNT originated from Hall strain of Clostridium botulinum [6], which made it possible to produce botulinum toxins and toxoid vaccines in large-scale. In the 1960's, when the treaty for biological weapon banned wartime uses of BoNTs, Dr. Ed-ward J. Schantz, a basic scientist who had worked at Camp Detrick, supplied the toxin for scientific purposes [7]. Dr. Alan B. Scott, an opthalmologist, came across an idea of its clin-ical uses at small doses to reduce muscle hyperactivity. He first tested botulinum toxin type A (BoNT/A) from the Hall strain from Dr Schantz in humans in 1978, after he received permission from the FDA. Ten years later, Allergan Inc. acquired the rights to distribute the drug or onabotulinumtoxinA (BOTOX®). The company extended clinical researches to obtain FDA approvals for its use in dystonia, spasticity, migraine and others.

Botulinum neurotoxins (BoNTs) are categorized into immunologically distinct 7 serotypes BoNT/A to /G. BoNT/A, which exist in various molecular weights from LL (900kD), L (500kD), M (300kD), and S (150kD). LL and L toxins contain hemagglutinin (HA) component and NTNH (non-toxin, non-hemagglutinin) component, which are not essential for the action of neurotoxin (S toxin), although LL toxin may have advantages over S because of its less diffusibility due to larger molecular weight[8]. M toxin is com-posed of NTNH and S toxin. Neurotoxin or S toxin comprises the heavy chain (HC), which has binding and translocation domains, and the light chain (LC), having a catalytic do-main.

Detailed researches on amino acid sequences on each serotype revealed subtypes among type A toxins (A1, A2) [9]. These subtypes have been increasingly recognized on each serotype. By now, serotype A is divided into subtypes A1-A8, and only A1 toxin from the Hall strain (OnabotulinumtoxinA or BOTOX®, AbobotulinumtoxinA or Dysport®, IncobotulinumtoxinA or Xeomin®) have been clinically available in US [10], except for type B toxin (RimabotulinumtoxinB or Myobloc®/ Neurobloc®). Type F was once used for clinical researches, but was found to have shorter duration of action than type A [11].

Please add multiple references at the end of each these sentences.

You said: Sakaguchi and colleagues [12] have reported unique strains of C botulinum obtained from cases of infant botulism in Japan, and found the strains (Chiba-H and Kyoto-F) only producing M-sized type A toxins, which later were categorized as subtype A2. Because of the high yield and purity of the M toxin produced by Chiba-H strain, BoNT/A2 preparation was easily cleaved and converted into highly purified S toxin (neurotoxin) for clinical uses (A2NTX). The latter was found to be less diffusible and more efficacious per mouse LD50 unit in vitro and in vivo. The first-in-man clinical study indicated that A2NTX is around 1.54 times as potent as the same unit of A1LL [13].

Please describe all these studies very exactly.

Results

2.1. Patients

Please add to this section the exactly information about the permission of all your experiments:

The organisation name, date and the protocol number of the permission for all your experiments.   

You said: Those who had A1LL, A1NTX, and A2NTX were diagnosed as having spasticity or dystonia/tremor. Total number of patients for each injection group were 90 (34 female 56 male) for A1LL, 89 (23 female 66 male) for A1NTX, and 120 for A2NTX (53 female 67 male) (Table 1), which included patients with spasticity (21 for A1LL, 29 for A1NTX and 42 for A2NTX) and dystonia (69 for A1LL, 60 for A1NTX and 78 for A2NTX). The breakdown of spasticity patients was post-stroke (17 for A1LL, 10 for A1NTX and 25 for A2NTX), hereditary spastic paraplegia (HSP) (2 for A1LL, 11 for A1NTX and 12 for A2NTX), spinal cord injuries (1 for A1LL, 4 for A1NTX and 4 for A2NTX), and others including cerebral palsy.

Please describe all these abbreviations used.

Table 1 and Sections 2.2, 2.3, 2.4. and 2.5.: please say exactly how you analysed all these side effects? Did you analysed all these side effects by yourself, other physician, or asking the patients?   Please describe this very important point in your manuscript.

Table 2: please say exactly how you collected all these date? Please describe this very important point in your manuscript.

You said: 2.6. Secondary Non-responders to A1LL

Those with resistance to A1LL tested by frontalis test..

Please add some information and appropriate references for frontal test.

Discussion

You said: The present study showed that A2NTX was well tolerable up to the dose of 500 units for spasticity and dystonia. As expected from the previous in vitro and in vivo stud-ies[12,15-19], the spread of the toxin was significantly less than A1NTX. It was even less than those who had smaller doses of A1LL. The margin of safety would be even higher considering that 1 unit of A2 is 1.54 times as effective as A1[13]. Although this study was not designed as testing the efficacies, A2NTX seemed to be more efficacious than A1 toxins with regards to the patients’ predilection, and was useful even in secondary non-respond-ers to A1LL if used at relatively high doses of A2.

Please describe all these studies very exactly.

You said: Concerning the treatment-related side effects (Table 1), A2NTX seems to be less spreading not only in animals[16,17,18], but also in clinical settings. Interestingly, a simi-larly prepared A1NTX showed more spreading to neighbouring or distant muscles than A2NTX or A1LL (Table 1). The large molecular weight of A1LL is possibly related to its immunogenicity, resulting in antibody development, but could be beneficial in limiting its diffusion to other muscles locally or through CNS[8]. Despite its low molecular weight (150kD), A2NTX seems to be less spreading not only than A1NTX but also A1LL in the present study. Our previous report also suggested its less spreading than A1LL in lower limb post-stroke spasticity[14]. The exact reason is unknown, but it may be related to its higher affinity and faster binding to the receptor SV2 possibly because of the difference of the amino acid residues in the heavy chain[19]. If so, it is reasonable to observe its efficacy in secondary non-responders to A1LL, who could have antibodies to A1 toxins, since A2

Please describe all these studies very exactly. All your arguments sounded only like “seems to be”, but like that is not “real” scientific grounds.

Conclusion

You said: The present study underscored the safety and tolerability of A2NTX, a low molecular weight neurotoxin, up to 500 mouse LD50 units per 10-12 weeks, in the long-term condi-tion without eliciting antibodies. A2 is not suited for its use as a biological weapon like A1 because of its less diffusibility or spread, but is a promising therapeutic agent in the clinical settings where the reduction in tone can be expected in targeted muscles.

All you arguments is only your personal opinion, but without scientific justification.

Figure 1 and Figure 2: please add to both “real” Legends.

5.5 Adverse Events

Please describe very exactly how did you analysed this?

Author Response

We appreciate the reviewer for helpful comments, and revised the manuscript accordingly as follows:

General points:
Please correct all spaces between the words and references numbers at the end of the sentences.

-We carefully revised the spaces between the words and the reference number in all the manuscript.

Please add to your manuscript List of Abbreviations.

-We added List of Abbreviations at p.1.

Please change your title to: Clinical Safety and Tolerability ofA2NTX, a Novel Low Molecular Weight Neurotoxin Derived from Botulinum Neurotoxin Sub-type A2, In Comparison with SubtypeA1 Toxins.

- We changed the title accordingly (p.1).

Special points:
My big problem with this study that this Neurotoxin Sub-type A2is not authorized for patients, but the authors did these experiments? Please describe very exactly in which countries is this medication authorized and possible?

- We conducted the study in Japan with approval of IRB at Tokushima University, Japan. A part of the study was registered (ClinicalTrials.gov identifier: NCT01910363). The study was legitimate in Japan. We added these points at the last para of Introduction (para 1, p.4).

Is this study an official first human study for this medication?

- No, the first-in-man was already published (ref. 21; Mukai et al. ).

All you arguments is only your personal opinion, but without scientific justification. Please describe very exactly how you did analysing of the efficiency and side effects in patients?

-We did not intend to analyze the efficacy in this paper. We analyzed the side effects according to ICH E2A (Clinical Safety Data Management: Definitions and Standards for Expedited Reporting) [26].

This was added as 2nd para of 2.2 (p.5) as follows: The patients were inquired for any adverse events at each visit by a nurse or clinical research coordinator (CRC) blinded as to the dose and the preparation of BoNT (A1LL, A1NTX, A2NTX), and physicians examined muscle power manually at each visit not only of the injected, but also un-injected muscles. The results were analyzed according to ICH E2A (Clinical Safety Data Management: Definitions and Standards for Expedited Reporting) [26] and categorized as follows: 

Keywords:
please add also to keywords: patients

We added at p.1

Introduction
First of all, please describe in your Introduction section: is this Neurotoxin Sub-type A2 authorised for patient’s treatment? In which countries exactly is this Neurotoxin Sub-type A2authorized for patients?

- We conducted the study in Japan with approval of IRB at Tokushima University, Japan. A part of the study was registered (ClinicalTrials.gov identifier: NCT01910363). The study was legitimate in Japan. We added these points at the last para of Introduction (para 1, p.4).

You said: Botulinum neurotoxins (BoNT) are known as the most potent biological substance to risk people to death. Many attempts to use BoNT as a biological weapon were made in the past.
Please add multiple references at the end of this sentence.

We added ref 1-3 at para 1, p.3.

You said: In US Army, investigators of Camp Detrick conducted researches on BoNT from 1943 to 1956 to develop toxins and effective vaccines during and after World War II. The isolation and purification for BoNT originated from Hall strain of Clostridium botulinum[6], which made it possible to produce botulinum toxins and toxoid vaccines in large-scale. In the1960's, when the treaty for biological weapon banned wartimeuses of BoNTs, Dr. Ed-ward J. Schantz, a basic scientist who had worked at Camp Detrick, supplied the toxin for scientific purposes [7]. Dr. Alan B. Scott, an opthalmologist, came across an idea of its clinical uses at small doses to reduce muscle hyperactivity. He first tested botulinum toxin
type A (BoNT/A) from the Hall strain from Dr Schantz in humans in 1978, after he received permission from the FDA. Ten years later, Allergan Inc. acquired the rights to distribute the drug oronabotulinumtoxinA (BOTOX®). The company extended clinical researches to obtain FDA approvals for its use in dystonia, spasticity, migraine and others.
Botulinum neurotoxins (BoNTs) are categorized into immunologically distinct 7 serotypes BoNT/A to /G. BoNT/A, which exist in various molecular weights from LL (900kD), L(500kD), M (300kD), and S (150kD). LL and L toxins contain hemagglutinin (HA) component and NTNH (non-toxin, non-hemagglutinin) component, which are not essential for the action of neurotoxin (S toxin), although LL toxin may have advantages over S because of its less diffusibility due to larger molecular weight[8]. M toxin is composed of NTNH and S toxin. Neurotoxin or S toxin comprises the heavy chain (HC), which has binding and translocation domains, and the light chain (LC),having a catalytic domain.
Detailed researches on amino acid sequences on each serotype revealed subtypes among type A toxins (A1, A2) [9]. These subtypes have been increasingly recognized on each serotype. By now serotypeA is divided into subtypesA1-A8 and only A1toxin from the Hall strain (OnabotulinumtoxinA or BOTOX®, AbobotulinumtoxinA or Dysport®, IncobotulinumtoxinA or Xeomin®) have been clinically available in US [10], except for type B toxin (RimabotulinumtoxinB or Myobloc®/ Neurobloc®). Type F was once used for clinical researches, but was found to have shorter duration of action than type A [11].

Please add multiple references at the end of each these sentences.

We added ref. 4-14 at para 1 -3, p.3.

You said: Sakaguchi and colleagues [12] have reported unique strains of C botulinum obtained from cases of infant botulism in Japan, and found the strains (Chiba-H and Kyoto-F) only producing M-sized type A toxins, which later were categorized as subtype A2. Because of the high yield and purity of the M toxin produced by Chiba-H strain, BoNT/A2 preparation was easily cleaved and converted into highly purified S toxin (neurotoxin)for clinical uses (A2NTX). The latter was found to be less diffusible and more efficacious per mouse LD50 unit in vitro and in vivo. The first-in-man clinical study indicated that A2NTX is around 1.54 times as potent as the same unit of A1LL [13].
Please describe all these studies very exactly.

-We expanded the description (para 4-5, p.3), and increased the references (ref 11-22). 

Results
2.1. Patients
Please add to this section the exactly information about the permission of all your experiments: The organisation name, date and the protocol number of the permission for all your experiments.

- We added these points at para 1, p.4. 

You said: Those who had A1LL, A1NTX, and A2NTX were diagnosed as having spasticity or dystonia/tremor. Total number of patients for each injection group were 90 (34 female 56 male)for A1LL, 89 (23 female 66 male) for A1NTX, and 120 for A2NTX(53 female 67 male) (Table 1), which included patients with spasticity (21 for A1LL, 29 for A1NTX and 42 for A2NTX) and dystonia (69 for A1LL, 60 for A1NTX and 78 for A2NTX). The breakdown of spasticity patients was post-stroke (17 for A1LL,10 for A1NTX and 25 for A2NTX), hereditary spastic paraplegia (HSP) (2 for A1LL, 11 for A1NTX and 12 for A2NTX), spinal cord injuries (1 for A1LL, 4 for A1NTX and 4 for A2NTX), and others including cerebral palsy.
Please describe all these abbreviations used.

- We described all these abbreviations at para 2, p.4 and List of Abbreviations at p.1. 

Table 1 and Sections 2.2, 2.3, 2.4. and 2.5.: please say exactly how you analysed all these side effects? Did you analysed all these side effects by yourself, other physician, or asking the patients? Please describe this very important point in your manuscript.

-As mentioned above, this was added as 2nd para of 2.2 (p.5) : The patients were inquired for any adverse events at each visit by a nurse or clinical research coordinator (CRC) blinded as to the dose and the preparation of BoNT (A1LL, A1NTX, A2NTX), and physicians examined muscle power manually at each visit not only of the injected, but also un-injected muscles. The results were analyzed according to ICH E2A (Clinical Safety Data Management: Definitions and Standards for Expedited Reporting) [26] and categorized as follows: 

Table 2: please say exactly how you collected all these date? Please describe this very important point in your manuscript.

-We added these points at para4-5 of section 2.2 (p.5):

When serious or any adverse events in question arose, the independent committee for safety was called, and members were asked to assess whether the event was causal or not.

Since all the complete medical records of A1LL- or A1NTX-treated groups were not available due to their observation period (A1NTX: 2006-2010) or referral from other medical institutions (A1LL), a thorough review of any adverse events were limited for A1 toxin recipients, and the bias is was in favor of A1.

and last para, p.6:

Table 2 depicts the rest of adverse events in A2NTX-treated group, picked up from all the medical records for more than 8 years (2010-2018) by a CRC or a physician blinded as to the dosing schedule. Falls were reported in 3 cases. One (case 50) had a hip bone fracture at the peak effect of A2NTX injected into the leg (4 weeks after the injection), which was judged as causal, because the patient became ambulatory and had a chance to walk for a long distance after a period of inactivity. This patient recovered completely after the joint replacement. There were 2 deaths (all with generalized dystonia), which were judged not directly related to the injection, one from a suicide and the other from an accident. Two patients had relapses of stroke not immediately following the injection (>2 weeks).

You said:
2.6. Secondary Non-responders to A1LL
Those with resistance to A1LL tested by frontalis test..
Please add some information and appropriate references for frontal test.

-We added ref 25 for frontalis test, and added the explanation of the test at 5.5 (p.11). 

Discussion
You said: The present study showed that A2NTX was well tolerable up to the dose of 500 units for spasticity and dystonia. As expected from the previous in vitro and in vivo studies[12,15-19], the spread of the toxin was significantly less than A1NTX. It was even less than those who had smaller doses of A1LL. The margin of safety would be even higher considering that 1 unit ofA2 is 1.54 times as effective as A1[13]. Although this study was not designed as testing the efficacies, A2NTX seemed to be more efficacious than A1 toxins with regards to the patients’ predilection, and was useful even in secondary non-responders to A1LL if used at relatively high doses of A2.
Please describe all these studies very exactly.

- We agree that this part of Discussion is too much speculative, and revised this as follows:

The present study was the first to report the long-term tolerability and safety of A2NTX, and showed that it was well tolerated up to the dose of 500 units for spasticity and dystonia. The limitation was that it was a retrospective case-control study not designed for statistical comparison between the preparations.

As expected from the previous in vitro and in vivo studies [17,18,27-30], the adverse events ascribed to the spread of the toxin were less frequent than A1NTX. Although the study was not designed to compare the incidence statistically, it was even less than those who had smaller doses of A1LL. If so, the margins of safety would be even higher considering that 1 unit of A2 is 1.54 times as effective as A1 [21].

With regards to the patients’ predilection, A2NTX was more favored than A1 toxins and was useful even in secondary non-responders to A1LL if used at relatively high doses of A2.

You said: Concerning the treatment-related side effects (Table1), A2NTX seems to be less spreading not only in animals [16,17,18], but also in clinical settings. Interestingly, a similarly prepared A1NTX showed more spreading to neighbouring or distant muscles than A2NTX or A1LL (Table 1).The large molecular weight of A1LL is possibly related to its immunogenicity, resulting in antibody development, but could be beneficial in limiting its diffusion to other muscles locally or through CNS[8]. Despite its low molecular weight (150kD), A2NTX seems to be less spreading not only than A1NTX but also A1LL in the present study. Our previous report also suggested its less spreading than A1LL in lower limb post-stroke spasticity[14]. The exact reason is unknown, but it may be related to its higher affinity and faster binding to the receptor SV2 possibly because of the difference of the amino acid residues in the heavy chain[19]. If so, it is reasonable to observe its efficacy in secondary non-responders to A1LL, who could have antibodies to A1 toxins, since A2

Please describe all these studies very exactly. All your arguments sounded only like “seems to be”, but like that is not “real” scientific grounds.

-We agree with the reviewer on this point, and carefully avoided 'seem to' wording (last para of Discussion, p.7) with increased number of references. 

Conclusion
You said: The present study underscored the safety and tolerability of A2NTX, a low molecular weight neurotoxin, up to500 mouse LD50 units per 10-12 weeks, in the long-term condition without eliciting antibodies. A2 is not suited for its use as a biological weapon like A1 because of its less diffusibility or spread, but is a promising therapeutic agent in the clinical settings where the reduction in tone can be expected in targeted muscles.
All you arguments is only your personal opinion, but without scientific justification.

- We added 'Although the present retrospective case-control study was not designed for statistical comparison between the preparations,' at the head of conclusion. We deleted 'A2 may not...' at Conclusion. (p.8). 

Figure 1 and Figure 2: please add to both “real” Legends.

-We added real legends to Fig.1 (p.9) and Fig.2 (p.10). 

5.5 Adverse Events
Please describe very exactly how did you analysed this?

- As mentioned above, we added detailed account on this at para1-2 of section 2.2, and added ref 25 for frontalis test. 

Round 2

Reviewer 2 Report

Dear authors,

Thank you for corrections. Unfortunately, this manuscript needs still some improvements and corrections before publishing may be possible.

Introduction

You said: Botulinum neurotoxins (BoNT) are known as the most potent biological substance to risk people to death. Many attempts to use BoNT as a biological weapon were made in the past [1-3]. In US Army, investigators of Camp Detrick conducted researches on BoNT from 1943 to 1956 to develop toxins and effective vaccines during and after World War II [4]. The isolation and purification for BoNT originated from Hall strain of Clostridium bot-ulinum [5][6], which made it possible to produce botulinum toxins and toxoid vaccines in large-scale [4]. In the 1960's, when the treaty for biological weapon banned wartime uses of BoNTs, Dr. Edward J. Schantz, a basic scientist who had worked at Camp Detrick, sup-plied the toxin for scientific purposes [6][7]. Dr. Alan B. Scott [7], an opthalmologistoph-thalmologist, came across an idea of its clinical uses at small doses to reduce muscle hy-peractivity [7]. He first tested botulinum toxin type A (BoNT/A) from the Hall strain from provided by Dr Schantz in humans in 1978, after he received permission from the FDA. Ten years later, Allergan Inc. acquired the rights to distribute the drug or onabotuli-numtoxinA (BOTOX®). The company extended clinical researches to obtain FDA approv-als for its use in dystonia, spasticity, migraine and others [8].

Botulinum neurotoxins (BoNTs) are categorized into immunologically distinct 7 serotypes BoNT/A to /G. BoNT/A, which exist in various molecular weights from LL (900kD), L (500kD), M (300kD), and S (150kD) [9]. LL and L toxins contain hemagglutinin (HA) component and NTNH (non-toxin, non-hemagglutinin) component, which are not essential for the action of neurotoxin (S toxin), although LL toxin may have advantages over S because of its less diffusibility due to larger molecular weight [10][8]. M toxin is composed of NTNH and S toxin. Neurotoxin (NTX) or S toxin comprises the heavy chain (HC), which has binding and translocation domains, and the light chain (LC), having a catalytic domain.

Detailed researches on amino acid sequences on each serotype revealed subtypes among type A toxins (A1, A2) [11][9]. These subtypes have been increasingly recognized on each serotype. By now, serotype A is divided into subtypes A1-A8, and only A1 toxin from the Hall strain (OnabotulinumtoxinA or BOTOX®, AbobotulinumtoxinA or Dysport®, IncobotulinumtoxinA or Xeomin®) have been clinically available in US [12][10] , except for type B toxin (RimabotulinumtoxinB or Myobloc®/ Neurobloc®). Type F was once used for clinical researches, but was found to have shorter duration of action than type A [13][11]. There has however been safety concern of these subtype A1 neurotoxins in treating cerebral palsy children, because cases were reported culminating in deaths possibly due to their spread to respiratory muscles [14].

Once again, please add multiple references at the end of each these sentences.

You said: The first-in-man clinical study indicated that A2NTX is around 1.54 times as potent as the same unit of A1LL with less spread to adjacent muscles [21][13].. Small-sized double blind controlled study of head-to-head comparison of onabotulinumtoxinA and A2NTX in post-stroke spasticity indicated higher efficacy and safety of A2 [22].

Please describe both these studies exactly.

You said: As expected from the previous in vitro and in vivo studies [17,18,27-30][12,15-19], the adverse events ascribed to the spread of the toxin wereas significantly less frequent than A1NTX. It was even less than those who had smaller doses of A1LLAlthough the study was not designed to compare the incidence statistically, it was even less than those who had smaller doses of A1LL. If so, tThe marginmargins of safety would be even higher considering that 1 unit of A2 is 1.54 times as effective as A1 [21][13].

With regards to the patients’ predilection, Although this study was not designed as testing the efficacies, A2NTX was seemed to be more efficaciousmore favored than A1 toxins with regards to the patients’ predilection, and was useful even in secondary non-responders to A1LL if used at relatively high doses of A2.

Please describe all these studies exactly.

Author Response

We are grateful to the reviewer for kind and thoughtful points raised, and we corrected mx accordingly as follows.

Introduction

You said: Botulinum neurotoxins (BoNT) are known as the most potent biological substance to risk people to death. Many attempts to use BoNT as a biological weapon were made in the past [1-3]. In US Army, investigators of Camp Detrick conducted researches on BoNT from 1943 to 1956 to develop toxins and effective vaccines during and after World War II [4]. The isolation and purification for BoNT originated from Hall strain of Clostridium bot-ulinum [5][6], which made it possible to produce botulinum toxins and toxoid vaccines in large-scale [4]. In the 1960's, when the treaty for biological weapon banned wartime uses of BoNTs, Dr. Edward J. Schantz, a basic scientist who had worked at Camp Detrick, sup-plied the toxin for scientific purposes [6][7]. Dr. Alan B. Scott [7], an opthalmologistoph-thalmologist, came across an idea of its clinical uses at small doses to reduce muscle hy-peractivity [7]. He first tested botulinum toxin type A (BoNT/A) from the Hall strain from provided by Dr Schantz in humans in 1978, after he received permission from the FDA. Ten years later, Allergan Inc. acquired the rights to distribute the drug or onabotuli-numtoxinA (BOTOX®). The company extended clinical researches to obtain FDA approv-als for its use in dystonia, spasticity, migraine and others [8].

Botulinum neurotoxins (BoNTs) are categorized into immunologically distinct 7 serotypes BoNT/A to /G. BoNT/A, which exist in various molecular weights from LL (900kD), L (500kD), M (300kD), and S (150kD) [9]. LL and L toxins contain hemagglutinin (HA) component and NTNH (non-toxin, non-hemagglutinin) component, which are not essential for the action of neurotoxin (S toxin), although LL toxin may have advantages over S because of its less diffusibility due to larger molecular weight [10][8]. M toxin is composed of NTNH and S toxin. Neurotoxin (NTX) or S toxin comprises the heavy chain (HC), which has binding and translocation domains, and the light chain (LC), having a catalytic domain.

Detailed researches on amino acid sequences on each serotype revealed subtypes among type A toxins (A1, A2) [11][9]. These subtypes have been increasingly recognized on each serotype. By now, serotype A is divided into subtypes A1-A8, and only A1 toxin from the Hall strain (OnabotulinumtoxinA or BOTOX®, AbobotulinumtoxinA or Dysport®, IncobotulinumtoxinA or Xeomin®) have been clinically available in US [12][10] , except for type B toxin (RimabotulinumtoxinB or Myobloc®/ Neurobloc®). Type F was once used for clinical researches, but was found to have shorter duration of action than type A [13][11]. There has however been safety concern of these subtype A1 neurotoxins in treating cerebral palsy children, because cases were reported culminating in deaths possibly due to their spread to respiratory muscles [14].

Once again, please add multiple references at the end of each these sentences.

- We added multiple references for each sentences para 1, 2,3 p.3 of Introduction.

You said: The first-in-man clinical study indicated that A2NTX is around 1.54 times as potent as the same unit of A1LL with less spread to adjacent muscles [21][13].. Small-sized double blind controlled study of head-to-head comparison of onabotulinumtoxinA and A2NTX in post-stroke spasticity indicated higher efficacy and safety of A2 [22].

Please describe both these studies exactly.

 -We added detailed results of these studies. para5, p3 at Introduction.

You said: As expected from the previous in vitro and in vivo studies [17,18,27-30][12,15-19], the adverse events ascribed to the spread of the toxin wereas significantly less frequent than A1NTX. It was even less than those who had smaller doses of A1LLAlthough the study was not designed to compare the incidence statistically, it was even less than those who had smaller doses of A1LL. If so, tThe marginmargins of safety would be even higher considering that 1 unit of A2 is 1.54 times as effective as A1 [21][13].

With regards to the patients’ predilection, Although this study was not designed as testing the efficacies, A2NTX was seemed to be more efficaciousmore favored than A1 toxins with regards to the patients’ predilection, and was useful even in secondary non-responders to A1LL if used at relatively high doses of A2.

Please describe all these studies exactly.

- We elaborated these at para 2-5 of Discussion (last para of p.7 and para 1-3 of p.8.). 

Round 3

Reviewer 2 Report

Thank you.